# Karyopherin-β1 Regulates Radioresistance and Radiation-Increased Programmed Death-Ligand 1 Expression in Human Head and Neck Squamous Cell Carcinoma Cell Lines

**DOI:** 10.3390/cancers12040908

**Published:** 2020-04-08

**Authors:** Masaharu Hazawa, Hironori Yoshino, Yuta Nakagawa, Reina Shimizume, Keisuke Nitta, Yoshiaki Sato, Mariko Sato, Richard W. Wong, Ikuo Kashiwakura

**Affiliations:** 1Cell-Bionomics Research Unit, Institute for Frontier Science Initiative, Kanazawa University, Kanazawa, Ishikawa 920-1192, Japan; masaharu.akj@gmail.com (M.H.); rwong@staff.kanazawa-u.ac.jp (R.W.W.); 2Department of Radiation Science, Hirosaki University Graduate School of Health Sciences, Hirosaki, Aomori 036-8564, Japanikashi@hirosaki-u.ac.jp (I.K.); 3Department of Radiological Technology, Hirosaki University School of Health Sciences, Hirosaki, Aomori 036-8564, Japan; 4Department of Radiation Oncology, Hirosaki University Graduate School of Medicine, Hirosaki, Aomori 036-8563, Japan; s_mariko@hirosaki-u.ac.jp

**Keywords:** head and neck squamous cell carcinoma, karyopherin-β1, radiosensitization, apoptosis, puma, ΔNp63, programmed death-ligand 1

## Abstract

Nuclear transport receptors, such as karyopherin-β1 (KPNB1), play important roles in the nuclear-cytoplasmic transport of macromolecules. Recent evidence indicates the involvement of nuclear transport receptors in the progression of cancer, making these receptors promising targets for the treatment of cancer. Here, we investigated the anticancer effects of KPNB1 blockage or in combination with ionizing radiation on human head and neck squamous cell carcinoma (HNSCC). HNSCC cell line SAS and Ca9-22 cells were used in this study. Importazole, an inhibitor of KPNB1, or knockdown of KPNB1 by siRNA transfection were applied for the blockage of KPNB1 functions. The roles of KPNB1 on apoptosis induction and cell surface expression levels of programmed death-ligand 1 (PD-L1) in irradiated HNSCC cells were investigated. The major findings of this study are that (i) blockage of KPNB1 specifically enhanced the radiation-induced apoptosis and radiosensitivity of HNSCC cells; (ii) importazole elevated p53-upregulated modulator of apoptosis (PUMA) expression via blocking the nuclear import of SCC-specific oncogene ΔNp63 in HNSCC cells; and (iii) blockage of KPNB1 attenuated the upregulation of cell surface PD-L1 expression on irradiated HNSCC cells. Taken together, these results suggest that co-treatment with KPNB1 blockage and ionizing radiation is a promising strategy for the treatment of HNSCC.

## 1. Introduction

Head and neck squamous cell carcinoma (HNSCC) is a lethal malignancy arising from the regions of the head and neck, such as the pharynx and oral cavity. The cumulative incidence of HNSCC was approximately 890,000 new cases in 2018, and it now ranks as the seventh most common cancer types worldwide [1,2]. In spite of combined treatment involving surgery, radiation therapy, and chemotherapy, the five-year survival rate of HNSCC patients is low, and the prognosis is still poor [3]. Part of the reason is that resistance to chemo- and radiotherapy often occurs [4]. Furthermore, recent evidence indicates that ionizing radiation increases cell surface programmed death-ligand 1 (PD-L1) expression, which is an immune checkpoint molecule and negatively regulates anti-tumor immune responses [5,6,7]. In order to overcome these responses, the development of an effective strategy is desired for the treatment of HNSCC.

Upon genotoxic stresses, including radiation, apoptosis and PD-L1 expression are initiated from the nuclear events, such as DNA damage responses and transcription-mediated pro- and anti-apoptotic effects [8,9,10,11]. The Bcl-2 family of proteins play central roles in regulating apoptosis, where most of them maximize its activity through post-translational modification. Among the Bcl-2 family members, p53-upregulated modulator of apoptosis (PUMA) is the most potent killer, and its activity is exclusively controlled by transcriptional regulation [12]. Since all proteins regulating those nuclear events must enter the nucleus with assistance from nuclear transport receptors, a disturbance in the nuclear import process is considered as a therapeutic strategy to prevent tumor radiation responses.

Nuclear transport receptors, such as the karyopherin-α (KPNA) and karyopherin-β1 (KPNB1) family, selectively aid the shuttle of karyophilic proteins harboring nuclear localization signals (NLSs) through a nuclear pore complex [13,14,15]. Moreover, the KPNA family recognizes and binds to NLS-containing cargo. KPNB1 mediates the docking of KPNA/NLS-containing cargo complex to the nuclear pore complex, thereby facilitating nuclear entry of the target cargo. There are seven subtypes in the human KPNA family, and each individual KPNA is cargo-specific [15]. Recently, we reported KPNA4 as a HNSCC specifically amplified KPNA, which establishes mitogen-activated protein kinase activation through Ras-responsive element-binding protein 1-nuclear import [16]. Further, the high expression level of KPNA2 is involved in the progression of human breast tumor or human hepatocellular carcinoma [17,18]. Thus, while importance of cancer-specific alteration of KPNA subtype in cancer biology is recognized, therapeutic benefits from blocking KPNB1, the master nuclear transport receptor mediating all KPNA subtypes, remains to be investigated further.

A unifying feature of SCC is the high-level expression of TP63 in 30% of the cases, and the major isoform of p63 expressed in SCC is ΔNp63α [19,20,21,22]. ΔNp63α functions as a positive and negative transcriptional regulator of different gene subsets [23,24,25]. As a nuclear protein, ΔNp63α requires the aid of KPNB1 during nuclear traffic [26]. Moergel et al. previously reported that the high expression of p63 is related to the poor radiation response and prognosis of patients with HNSCC [27]; however, how p63 is involved in DNA damage response has not been extensively explained.

In this study, the authors investigated the effects of the KPNB1 inhibitor importazole (IPZ) or KPNB1 knockdown on the radiation response of HNSCC cells. Furthermore, it was shown that IPZ treatment enhanced the radiation-induced apoptosis and radiosensitivity of HNSCC cells, where relieving ΔNp63-mediated transcriptional silencing of PUMA expression is the central axis. The researchers also found that IPZ treatment or KPNB1 knockdown suppressed the radiation-induced upregulation of cell surface PD-L1 expression. In addition, the cytotoxic effect of IPZ was less in human umbilical vein endothelial cells (HUVEC) compared with HNSCC cells, and IPZ hardly enhanced the radiation-induced apoptosis in HUVEC. These findings highlight KPNB1 as a promising target to improve radiation therapy for HNSCC.

## 2. Results

### 2.1. High Expression Levels of KPNB1 Determines the Proliferation and Apoptosis of HNSCC Cells

We first profiled the clinical status of KPNB1 in HNSCC by re-analyzing data from The Cancer Genome Atlas (TCGA) cohorts via Cancer RNA-Seq Nexus or cBioportal for Cancer Genomics (see Section 4.13 for URLs). As shown in Figure 1A, TCGA RNA-seq data of the HNSCC cohort revealed that the expression level of KPNB1 in tumor tissue was significantly higher than that of normal tissue. In addition, the Kaplan–Meier analysis on the TCGA cohorts showed that HNSCC patients with high expressions of KPNB1 resulted in a poor outcome (Figure 1B). These findings imply that KPNB1 is related to the malignancy of HNSCC; therefore, we next examined whether KPNB1 is functionally involved in the proliferation as well as the viability of HNSCC cell lines (SAS and Ca9-22) using the knockdown of KPNB1 by siRNA transfection or the KPNB1 inhibitor IPZ. The silencing of KPNB1 by siRNA decreased the clonogenic potential to about 3% for SAS cells and 36% for Ca9-22 cells, respectively (Figure 1C,D), and enhanced the apoptosis of both SAS and Ca9-22 cells (Figure 1E). Furthermore, the dose-dependent treatment with IPZ decreased the viable cell number of SAS and Ca9-22 cells (Appendix A). The 50% inhibitory concentrations (IC_50_) of IPZ were 2.9 μM for SAS cells and 0.9 μM for Ca9-22 cells (Appendix A). IPZ decreased the clonogenic potential and enhanced the apoptosis of HNSCC cells (Figure 1F,G). Taken together, these results indicate that KPNB1 is involved in the proliferation of HNSCC cells.

### 2.2. KPNB1 Regulates Radioresistance of HNSCC Cells

The apoptotic induction rate determines the efficacy of radiation therapy. Apoptosis is evoked by two independent pathways, namely the extrinsic and intrinsic pathways. The extrinsic pathway causes caspase 8 activation while the latter depends on caspase 9 activation. As found in the study, the blockage of KPNB1 significantly augmented apoptosis, and the researchers investigated the pathway that was activated by either radiation alone, IPZ alone, or a combination of both. Western blot results showed that the presence of IPZ activated the caspase-9-mediated apoptosis pathway, which was not observed in radiation alone (Figure 2A). As a result of caspase 9 activation, the combination of IPZ and radiation markedly increased apoptotic cells as compared with either radiation or IPZ alone (Figure 2B,C). Similarly, KPNB1 depletion by siRNA also enhanced radiation-induced apoptotic cells (Figure 2D). Importantly, analysis of survival fraction demonstrated that IPZ has synergistic effects on the clonogenic survival of SAS and Ca9-22 cells with radiation treatment (Figure 2E, Appendix B). Collectively, these results indicate that caspase-9 activation by preventing KPNB1 functions is important for causing radiosensitizing effects on HNSCC cells.

### 2.3. IPZ Abolishes the Transcriptional Silencing of PUMA by Blocking p63 Nuclear Import in HNSCC Cells

Since PUMA is a major factor in activating caspase-9–mediated apoptosis [12], the researchers next determined whether or not PUMA is involved in apoptosis enhancement by IPZ. Previously, the authors, as well as other resources, reported that the lineage-specific oncogene ΔNp63, the major isoform expressed in SCCs, transcriptionally suppresses *BBC3* (PUMA) [26,28].

First, the authors profiled the occupancy of p63, as well as H3K27me (silencing mark for transcripts) around BBC3 gene, by reanalyzing ChIP-Seq data. The promoter regions were occupied by p63 together with H3K27me, indicating BBC3 is transcriptionally suppressed by p63 (Figure 3A). Next, the localization analysis of ΔNp63 was addressed by confocal imaging. It was observed that IPZ treatment caused diffusion or less nuclear localization of ΔNp63 in the HNSCC cell (Figure 3B and Appendix A). Lastly, IPZ treatment significantly increased PUMA expression at both mRNA and protein levels (Figure 3C,D), suggesting that IPZ enhances PUMA expression by preventing ΔNp63 nuclear import. Since KPNB1 blockage was reported to block c-JUN nuclear import [29], the localization of c-JUN was also investigated (Appendix A). Consistent with previous reports [29], the nuclear levels of c-JUN were inhibited by IPZ (Appendix A). Of utmost importance is the examination of TCGA RNA-seq datasets that further showed that the expression levels of PUMA are inversely related to the levels of p63 and not of the activator protein 1 (AP-1) family, including c-JUN (Figure 3E,F and Appendix A). Collectively, these results suggest that IPZ can trigger the elevation of PUMA expression via blocking ΔNp63 nuclear import in HNSCCs.

### 2.4. Radiosensitization Effects through Abolishing ΔNp63α-Mediated PUMA Silencing Is Unique to HNSCC Cells

Among several cancers (lung cancer, prostate cancer, and breast cancer) applied for radiation therapy, a poor prognosis in lung adenocarcinoma was observed in patients with high levels of KPNB1 (Appendix A). Since ΔNp63α is an SCC-specific oncogene, the researchers next investigated whether IPZ treatment confer radiosensitization effects on lung adenocarcinoma cells. Unsurprisingly, ΔNp63 expression was observed in only HNSCC cells, while the expression level of KPNB1 was noted in both HNSCC and lung adenocarcinoma cells (Figure 4A). However, IPZ showed no effects on both the mRNA and protein levels of PUMA (Figure 4B,C). Furthermore, although the IC_50_ value of IPZ for A549 cells (3.2 μM) was similar to that for SAS cells (Appendix A), there was no significant effect of IPZ on radiation-induced apoptosis, as well as radiosenstivity, in A549 cells (Figure 4D,E). More importantly, the negative correlation between TP63 and BBC3 was only observed in SCCs samples, including HNSCC, whereas it was not observed in non-SCC samples, such as lung adenocarcinoma and hepatocellular carcinoma (Figure 4F). Similar to the results of A549 cells, IPZ failed to enhance the radiosensitivity of hepatocellular carcinoma cell line HepG2 lacking ΔNp63 expression (Appendix A). These results strongly emphasize the blockage of ΔNp63–mediated PUMA suppression as the key pathway involved in radiosensitization in irradiated HNSCC cells.

### 2.5. KPNB1 Is Involved in Radiation-Increased Cell Surface PD-L1 Expression on HNSCC Cells

PD-L1 (also known as CD274) is one of the ligands of the immunoinhibitory molecule PD-1 expressed on immune cells, including T cells [5], and the interaction of PD-L1 and PD-1 negatively regulates immune response. Therefore, cancer cells expressing PD-L1 on their cell surface may impede host immunity against cancer cells. Since ionizing radiation is known to increase PD-L1 expression of cancer cells [6,11], the authors then investigated the involvement of KPNB1 in the upregulation of cell surface PD-L1 expression on irradiated HNSCC cells. As shown in Figure 5A, the expression of PD-L1 on HNSCC cells increased upon 6 Gy X-ray irradiation. Intriguingly, the blockage of KPNB1 functions significantly suppressed the radiation-increased PD-L1 expression in irradiated HNSCC cells (Figure 5A,B). Similarly, KPNB1 knockdown suppressed the radiation-increased PD-L1 expression on HNSCC cells (Appendix A). While IPZ treatments blocked ΔNp63 nuclear import and induced PUMA expression, IPZ itself did not affect PD-L1 expression in non-irradiated HNSCC cells (Figure 5C). To get an insight on PD-L1 regulation, a correlational analysis between CD274 (PD-L1) mRNA levels and TP63 (ΔNp63), BBC3 (PUMA), and interferon regulatory factor 1 (IRF1) reported as PD-L1 inducible transcription factor in irradiated cells [11] was performed. We found that the mRNA levels of CD274 significantly showed a strong and positive correlation with IRF1 expression levels, which was not observed for either TP63 or BBC3 (Figure 5D,E). Furthermore, a positive correlation between CD274 mRNA and IRF1 mRNA was observed in both non-SCC and SCC samples (Figure 5F), suggesting that the regulation of PD-L1 expression depends on IRF1.

## 3. Discussion

Many reports, including the present study, have demonstrated an involvement of nuclear transport receptors in the progression of cancer, making them promising targets for the treatment of cancer [16,30]. Since radiation-chemotherapy is the standard approach for HNSCC treatment, it is extremely important to know whether the blockage of nuclear transport receptors has a therapeutic benefit in that axis. In this study, it was demonstrated that the combination of KPNB1 blockage and radiation stimuli synergistically enhanced apoptosis by upregulating pro-apoptotic PUMA in HNSCC cells. Furthermore, the radiosensitizing effects was unique to HNSCC, since PUMA regulation is under the control of the SCC-specific oncogene ΔNp63α. Note that the researchers further observed that IPZ treatment or KPNB1 knockdown attenuated the upregulation of cell surface PD-L1 expression on irradiated HNSCC cells, suggesting that a combination of KPNB1 blockage and radiation is highly expected as a novel therapeutic approach for HNSCC.

Radiation treatment is expected to cause apoptotic cell death by damaging the genomic DNA in cancer cells. Apoptosis is a physiological process activated by caspase-dependent protease cascades [31]. Caspase activation depends on two pathways, namely the extrinsic and intrinsic pathways. The authors found that either IPZ or its combination with radiation enhanced the intrinsic pathway, as characterized by caspase-9 activation. There was a further highlight on PUMA as the key pro-apoptotic factor contributing to the radiosensitization effects of IPZ in irradiated HNSCC cells, which is consistent with the results of previous studies that demonstrated PUMA expression evokes a profound chemo- and radiosensitization in several cancer cells [32,33,34]. While the DNA repair process requires the karyopherin-dependent import of DNA repair proteins, such as BRAC1, NBS1, and p53 binding protein 1 in cancer cells [8,35,36], it would be interesting to determine whether KPNB1 inhibition disturbs the importation such repair proteins in HNSCC cells.

Moreover, ΔNp63α was identified as the candidate target of KPNB1 in determining the radiosensitivity of HNSCC cells. Functionally, KPNB1-mediated ΔNp63α nuclear transport is crucial in suppressing pro-apoptotic PUMA expression and consequentially lowering the survival rate in irradiated HNSCC cells. TP63 gene, a p53 homolog, is a master transcriptional regulator of epithelial development and maintenance [37]. Furthermore, ΔNp63α is known as an oncogenic transcription factor in SCC [38]. Because ΔNp63α acts as the lineage-specific oncogene, the synergistic effects of IPZ and radiation were demonstrated in HNSCC cells but were not reported in human lung adenocarcinoma A549 cells, hepatocellular carcinoma HepG2 cells, or in normal human umbilical vein endothelial cells (HUVEC) (Appendix A). The cytotoxic effect of IPZ was less in HUVEC, and it hardly enhanced the radiation-induced toxicity on HUVEC (Appendix A). Therefore, the application of IPZ in chemo-radiotherapy has positive benefits in HNSCC. It would be noteworthy to investigate whether the blockage of KPNB1 regulates the cellular radiosensitivity of other SCCs in future studies.

Recent evidence shows that ionizing radiation increases PD-L1 expression in various cancer cells [39,40,41]. In this study, however, the researchers showed that blocking KPNB1 functions suppressed the induction of PD-L1 expression in irradiated HNSCC cells, suggesting that KPNB1’s nuclear transport-regulating action is involved in the radiation-increased PD-L1 expression. Consistent with the previous study [11], the transcript amounts of PD-L1 had a significant positive correlation to IRF1 levels but not with either TP63 or PUMA expression levels. IRF1 contains NLS [42] and is transported by KPNA2 upon interferon-γ stimuli [43]. Therefore, blocking the KPNA2-IRF1 nuclear import may serve as a promising mechanism to prevent abominable PD-L1 upregulation during radiation therapy.

## 4. Materials and Methods 

### 4.1. Bioinformatics and Data Analysis

KPNB1 expression in tumor and non-tumor tissue samples was downloaded from Cancer RNA-Seq Nexus (see Section 4.13 for URLs) and reanalyzed. The relationship between high KPNB1 expressions (mRNA expression z-Scores (RNA Seq V2 RSEM) > mean + 0.5 SD) and overall survival of different types of cancer patients in TCGA cohorts was analyzed through cBioportal for Cancer Genomics (see Section 4.13 for URLs). ChIP-seq profiles were reanalyzed based on Cistrome Data Browser (see Section 4.13 for URLs).

### 4.2. Reagents

IPZ, propidium iodide (PI), and dimethyl sulfoxide (DMSO) were purchased from Sigma-Aldrich (St. Louis, MO, USA). Anti-caspase-8 antibody (#9746), caspase-9 antibody (#9502), PUMA antibody (#12450), β-actin antibody (#4967), anti-rabbit IgG horseradish peroxidase (HRP)-linked antibody (#7074), and anti-mouse IgG HRP-linked antibody (#7076) were purchased from Cell Signaling Technology Japan, K.K. (Tokyo, Japan). The anti-importin beta mouse monoclonal antibody [3E9] (ab2811) was purchased from abcam (Cambridge, UK). The phycoerythrin (PE)-labeled anti-human PD-L1 monoclonal antibody (#329706), PE-conjugated anti-mouse IgG_2b_ antibody (#400314), and anti-p63(ΔN) antibody (#619001) were purchased from BioLegend (San Diego, CA, USA). Ambion Silencer^®^ Select Pre-designed siRNA against the gene-encoding KPNB1 (cat. no. s7919) and Silencer^®^ Select Negative #1 Control siRNA (cat. no. AM4611) were purchased from Thermo Fisher Scientific, Inc. (Waltham, MA, USA).

### 4.3. Cell Culture and Treatment

SAS, Ca9-22, and A549 cells were obtained from RIKEN Bio-Resource Center (Tsukuba, Japan). HepG2 and RERF-LC-MS cells were from the Japanese Collection of Research Bioresources Cell Bank (Osaka, Japan). SAS and Ca9-22 cells were maintained in high glucose Dulbecco’s modified eagle medium (DMEM; Wako Pure Chemical Industries, Ltd., Osaka, Japan) supplemented with 10% heat-inactivated fetal bovine serum (FBS, Japan Bioserum Co., Ltd., Fukuyuma, Japan) and 1% penicillin/streptomycin (P/S, Wako Pure Chemical Industries). A549 and HepG2 cells were maintained in low glucose DMEM (Sigma-Aldrich) supplemented with 1% P/S and 10% heat-inactivated FBS. RERF-LC-MS were maintained in Minimum Essential Medium Eagle (Sigma-Aldrich) supplemented with 1% P/S and 1% MEM non-essential amino acids (Thermo Fisher Scientific, Inc.), and 10% heat-inactivated FBS. H1299 cells from the American Type Culture Collection (ATCC, Manassas, VA, USA) were maintained in RPMI1640 medium (Gibco^®^; Invitrogen/Thermo Fisher Scientific, Waltham, MA, USA) supplemented with 1% P/S and 10% heat-inactivated FBS. HUVEC were purchased from Cell Applications, Inc. (San Diego, CA, USA) and maintained in an endothelial cell growth medium kit (D12023, Takara Bio Inc., Shiga, Japan) in type I collagen-coated dishes (IWAKI Glass Co. Ltd., Chiba, Japan). All cell lines were cultured at 37°C in a humidified atmosphere containing 5% CO_2_.

The cells were seeded in 12-well plates (5 × 10^4^ cells; BD Falcon; BD Biosciences, Franklin Lakes, NJ, USA), 35-mm culture dishes (10 × 10^4^ cells; IWAKI Glass Co. Ltd.), or 60-mm culture dishes (20 × 10^4^ cells; IWAKI Glass Co. Ltd.) and incubated for 6 h to allow adherence. After incubation, IPZ was added to the culture medium. DMSO-treated cells were prepared as vehicle controls. After 4 days of culturing at 37 °C, the cultured cells were harvested using 0.25% trypsin-ethylenediaminetetraacetic acid (Wako Pure Chemical Industries, Ltd.), and the number of viable cells was counted using trypan blue dye exclusion assay before subsequent analysis. In some experiments, X-ray irradiation was performed at 1 h after IPZ administration and the cells were incubated for 4 days or about 20 h in the presence of IPZ. After the 20 h-incubation, the cells were washed twice with fresh media, and then the culture medium was replaced. After the medium replacement, the cells were further cultured for 1–3 days at 37 °C and harvested for subsequent analyses.

### 4.4. siRNA Transfection

The SAS or Ca9-22 cells were transfected with siRNA, targeting either KPNB1 or Control siRNA, using Lipofectamine^®^ RNAiMAX (Invitrogen; Thermo Fisher Scientific, Inc.), following the manufacturer’s instructions. The final siRNA concentration was 5 nM. After transfection for 48 h, the cells were harvested and used for subsequent analyses.

### 4.5. In Vitro X-ray Irradiation

X-ray irradiation (150 kVp, 20 mA, 0.5 mm Al, and 0.3-mm Cu filters) was performed using an X-ray generator (MBR-1520R-3; Hitachi Medical Corporation, Tokyo, Japan) at a distance of 45 cm from the focus, with a dose rate of 0.99–1.04 Gy/min, which was monitored by placing a thimble ionization chamber next to the sample during irradiation.

### 4.6. Clonogenic Survival Assay

The cells were seeded in 6-well plates (BD Falcon) or 60-mm culture dishes (IWAKI Glass Co. Ltd.) and were incubated for 6 h to promote adherence to the dish. After incubation, 0.1% DMSO or IPZ was added to the culture medium and the cells were cultured for 8–12 days. To further investigate the radiosensitizing effect of IPZ, 0.1% DMSO or IPZ was added to the culture medium 1 h before irradiation and then exposed to X-rays. After X-ray irradiation, the cells were incubated for about 20 h in the presence of IPZ. Next, the cells were washed twice with fresh media and their culture medium was replaced with a fresh set. After the replacement, the cells were cultured for 8–12 days. Subsequently, the cells were fixed with methanol and were stained with Giemsa solution (Wako Pure Chemical Industries, Ltd.). The colonies that contained >50 cells were counted. The surviving fraction at each radiation dose was calculated based on previously reported procedures in this study [44].

### 4.7. Detection of Apoptosis

Apoptosis was analyzed by annexin V-FITC (BioLegend) and PI staining as previously reported [45]. Moreover, the cells treated with each compound were harvested, washed, and suspended in annexin V Binding Buffer (BioLegend). The annexin V-FITC (2.5 µg/mL) and PI solution (50 µg/mL) were added to the cell suspension and incubated for 15 min in the dark at room temperature. Then, the apoptotic cells were analyzed via flow cytometry (Cytomics FC500; Beckman–Coulter, Fullerton, CA, USA).

### 4.8. SDS-PAGE and Western Blotting

SDS-PAGE analysis and Western blotting were performed based on previously reported procedures in this study [46]. The primary antibodies that were used are anti-caspase-8 antibody (1:3000), caspase-9 antibody (1:3000), puma antibody (1:3000), p63 (ΔN) antibody (1:3000), importin beta antibody (1:3000), and β-actin antibody (1:4000). The following secondary antibodies that were used were HRP-linked anti-rabbit IgG antibody (1:10,000) or HRP-linked anti-mouse IgG antibody (1:10,000). The antigens were visualized using the Clarity^TM^ Western ECL Substrate (Bio-Rad Laboratories, Inc., Hercules, CA, USA). Blot stripping was performed using Stripping Solution (Wako Pure Chemical Industries, Ltd.).

### 4.9. Immunofluorescence Analysis

The cells on coverslips were incubated under the indicated conditions and fixed for 10 min in 4% paraformaldehyde in Ca2+- and Mg2+-free phosphate-buffered saline [PBS(−)] and then permeabilized with 0.3% Triton X-100 in PBS(−) for 3 min at room temperature. The coverslips were incubated with the indicated primary antibodies for 2 h. They were washed three times and incubated with Alexa Fluor-conjugated secondary antibodies (Life Technologies; Thermo Fisher Scientific, Waltham, MA, USA) for 1 h. After three washes, the samples were mounted onto coverslips using Pro-Long Gold Antifade reagent (Life Technologies) and examined by confocal microscopy (objective × 60/1.2, FluoView^®^ FV10i, Olympus, Tokyo, Japan).

### 4.10. Quantitative Reverse Transcription Polymerase Chain Reaction (qRT-PCR)

The total RNA extraction and the synthesis of complementary DNA templates were performed as previously indicated [47]. The synthesis of complementary DNA templates was performed using PrimeScript™ RT Master Mix (Takara Bio Inc.), used according to the manufacturer’s instructions, and quantitative RT–PCR was performed by SYBR^®^ Premix Ex Taq™ II (Takara Bio Inc.) in a Thermal Cycler Dice^®^ Real Time System (Takara Bio Inc.). The relative mRNA expression level of the target genes was calculated using GAPDH as a loading control. The primers that were used for this are listed in Table 1.

### 4.11. Analysis of Cell Surface PD-L1 Expression

The analysis of the cell surface PD-L1 expression was performed. In brief, harvested cells were washed twice with PBS(−) and then stained with PE-conjugated anti-human PD-L1 antibody or PE-conjugated anti-mouse IgG_2b_ isotype control for 30 min at 4 °C in the dark. After staining, the cells were washed and analyzed using flow cytometry.

### 4.12. Statistical Analysis

Data are presented as mean ± SD. The comparisons between the control and experimental groups were performed using the two-sided Student’s *t*-test or Mann–Whitney U-test depending on data distribution. Multiple batches of data were analyzed using one-way analysis of variance, followed by the Tukey–Kramer test. The differences were considered significant when the resulting *p* < 0.05. The Excel 2016 software (Microsoft, Washington, DC, USA), with the add-in software Statcel 4 (The Publisher OMS Ltd., Tokyo, Japan), was used to perform these statistical analyses. A sample *t*-test was performed using GraphPad QuickCalcs (see URLs), and the control group was considered as 100%.

### 4.13. URLs

Cancer RNA-Seq Nexus, http://syslab4.nchu.edu.tw/; cBioportal for Cancer Genomics, http://www.cbioportal.org/; Cistrome Data Browser, http://cistrome.org/db/#/; GraphPad QuickCalcs, http://graphpad.com/quickcalcs/OneSampleT1.cfm

## 5. Conclusions

In conclusion, the results in the study showed the beneficial effects of KPNB1 blockage on radiation response in HNSCC cells in terms of radiosensitization and inhibition of upregulation of cell surface PD-L1 expression. Therefore, co-treatment with KPNB1 blockage and ionizing radiation is a promising strategy for HNSCC therapy.

## Figures and Tables

**Figure 1 cancers-12-00908-f001:**
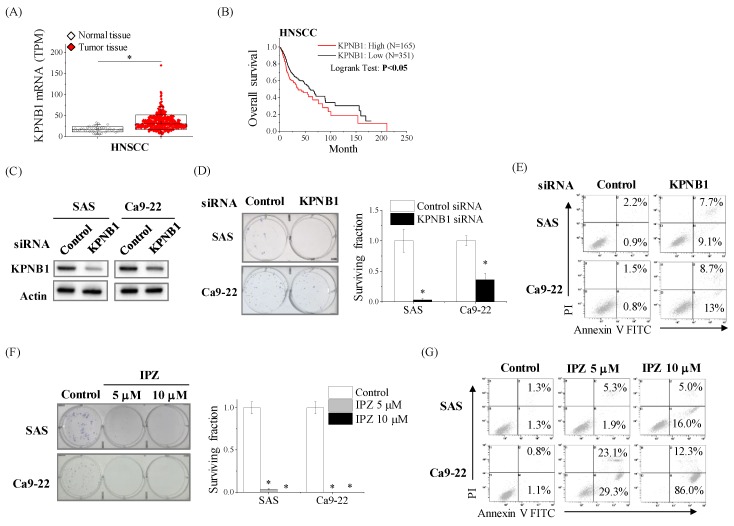
High expression levels of karyopherin-β1 (KPNB1) determines the proliferation, as well as the apoptosis resistance, of head and neck squamous cell carcinoma (HNSCC) cells. (**A**) The expression of KPNB1 in non-tumor tissue and HNSCC samples from Cancer RNA-Seq Nexus are shown. * indicates *p* < 0.01. (**B**) The relationship between the overall survival and KPNB1 expression of HNSCC patients in the The Cancer Genome Atlas (TCGA) cohorts is shown. The patients whose KPNB1 mRNA expression z-Scores (RNA Seq V2 RSEM) is greater than 0.5 SD above mean were defined as KPNB1 high patients. (**C**) HNSCC cells (SAS and Ca9-22 cells) transfected with control or KPNB1 siRNA were harvested, and KPNB1 protein expressions were analyzed by Western blot. The representative image of immunoblot is shown. Actin was used as loading control. (**D**) The clonogenic potential of HNSCC cells transfected with control or KPNB1 siRNA was estimated by colony formation assay. (Left) The representative pictures of the colonies are shown. (Right) The number of colonies from the cells transfected with control siRNA is considered 1.0. Data are presented as the mean ± SD of at least three independent experiments. * *p* < 0.05 versus control siRNA. (**E**) HNSCC cells transfected with control or KPNB1 siRNA were cultured for 4 days, and harvested for apoptosis assay using annexin V/propidium iodide (PI) staining. The representative cytograms of annexin V/PI are shown. The inset numbers indicate the proportion of annexin V^+^/PI^−^ cells or annexin V^+^/PI^+^ cells. (**F**) (Left) The representative pictures of the colony of HNSCC cells cultured in the presence of dimethyl sulfoxide (DMSO) or importazole (IPZ) are shown. (Right) The number of colonies from the cells treated with DMSO control is considered 1.0. Data are presented as the mean ± SD of at least three independent experiments. * *p* < 0.05 versus DMSO. (**G**) HNSCC cells cultured in the presence of IPZ for 4 days were harvested for apoptosis assay using annexin V/PI staining. The representative cytograms are shown. Inset numbers indicate the proportion of annexin V^+^/PI^−^ cells or annexin V^+^/PI^+^ cells.

**Figure 2 cancers-12-00908-f002:**
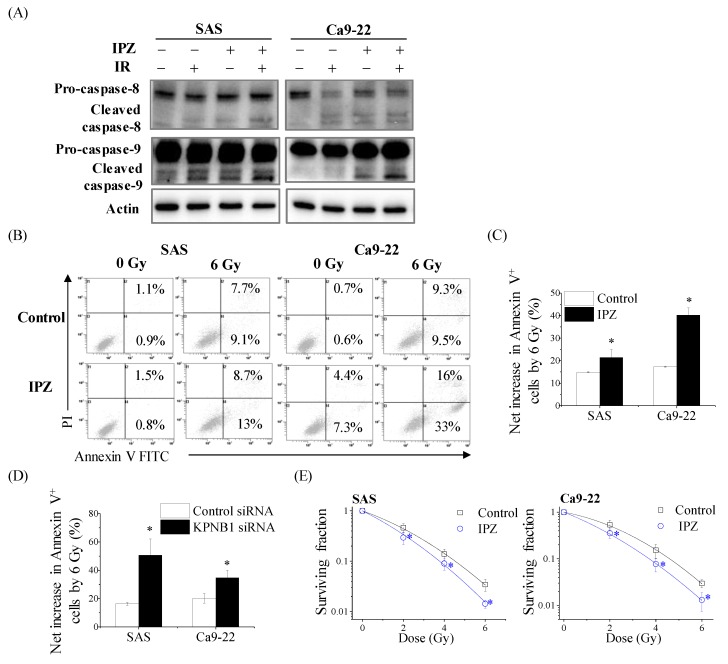
IPZ enhances the radiation-induced apoptosis and radiosensitivity of HNSCC cells. (**A**) IPZ (10 μM) was added to the culture medium 1 h before X-ray irradiation. After 20 h-incubation, the cell-culture-conditioned medium was replaced with fresh media and the cells were further cultured for 24 h. The cultured SAS and Ca9-22 cells were harvested for Western blot analyses of caspase-8 and -9. The representative immunoblots are shown. Actin was used as loading control. (**B**,**C**) IPZ (10 μM) was added to the culture medium 1 h before X-ray irradiation. After 20 h-incubation, the cell-culture-conditioned medium was replaced with fresh media and the cells were further cultured for 3 days. After culturing, the SAS and Ca9-22 cells were harvested for apoptosis assay using annexin V/PI staining. (**B**) The representative cytograms are shown. Inset numbers indicate the proportion of annexin V^+^/PI^−^ cells or annexin V^+^/PI^+^ cells. (**C**) The results are presented as the net increase in the population of annexin V^+^ cells (the sum of annexin V^+^/PI^−^ cells or annexin V^+^/PI^+^ cells) by 6 Gy irradiation. Data are presented as the mean ± SD of three independent experiments. * *p* < 0.05 versus DMSO control. (**D**) SAS and Ca9-22 cells transfected with control or KPNB1 siRNA were irradiated with 6 Gy X-ray and cultured for 4 days. After culturing, the cells were harvested for apoptosis assay using annexin V/PI staining. The results are presented as the net increase in the population of annexin V^+^ cells (the sum of annexin V^+^/PI^−^ cells or annexin V^+^/PI^+^ cells) by 6 Gy irradiation. Data are presented as the mean ± SD of three independent experiments. * *p* < 0.05 versus control siRNA. (**E**) IPZ (10 μM) was added to the culture medium 1 h before X-ray irradiation. After 20 h-incubation, the cell-culture-conditioned medium was replaced with fresh media and the cells were further cultured until the colony was observed. The surviving fraction of SAS and Ca9-22 cells is shown. Data are presented as the mean ± SD of three independent experiments performed in triplicate. * *p* < 0.01 versus DMSO.

**Figure 3 cancers-12-00908-f003:**
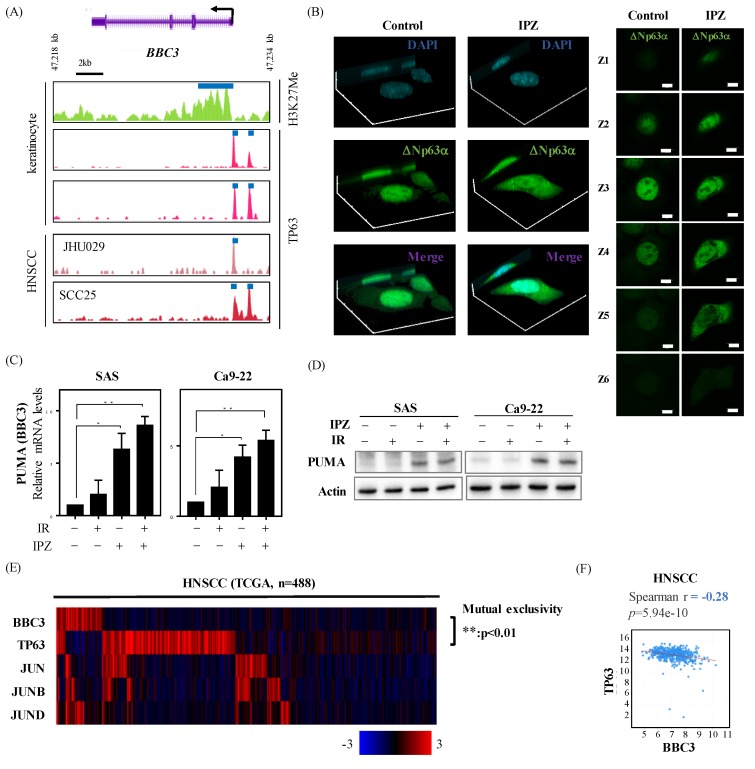
IPZ silences pro-apoptotic p53-upregulated modulator of apoptosis (PUMA) by suppressing the ΔNp63a nuclear import in HNSCC cells. (**A**) The occupancy profiles of H3K27me and TP63 at the BBC3 (PUMA) promoter in keratinocytes and HNSCCs. (**B**) Confocal imaging of ΔNp63α in SAS cells. Three-dimensional construction with maximum intensity (Left), and sequential z-stack imaging (Right), respectively. (**C**,**D**) IPZ (10 μM) was added to the culture medium 1 h before X-ray irradiation. After 20 h-incubation, the cells were harvested for qRT-PCR and Western blot analyses. (**C**) qRT-PCR analysis of PUMA expression levels. Data are presented as the mean ± SD of three independent experiments performed in triplicate. * *p* < 0.05, ** *p* < 0.01 versus DMSO control. (**D**) Western blot analysis of PUMA expression. The representative images are shown. (**E**) The heat map showing mutual exclusivity between BBC3 (PUMA) expression and TP63, as well as activator protein 1 (AP-1) family genes. The samples were divided according to mRNA expression levels (mRNA expression z-Scores (RNA-Seq V2 RSEM) > mean + 1.0 SD) from the TCGA cohorts. The P values are based on Fisher’s exact test. (**F**) The correlation between each BBC3 (PUMA) mRNA and TP63 mRNA in HNSCC from TCGA.

**Figure 4 cancers-12-00908-f004:**
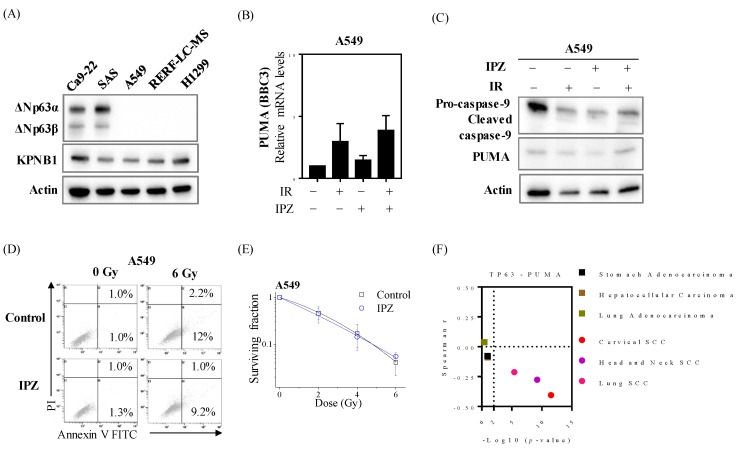
Radiosensitization effects through abolishing ΔNp63α-mediated PUMA silencing is unique to HNSCC cells. (**A**) Western blot analysis of ΔNp63 and KPNB1 expression. The representative image is shown. (**B**,**C**) IPZ (10 μM) was added to the culture medium 1 h before X-ray irradiation. After 20 h-incubation, the cells were harvested for qRT-PCR and Western blot analyses. (**B**) qRT-PCR analysis of PUMA expression levels. Data are presented as the mean ± SD of three independent experiments performed in triplicate. (**C**) Western blot analysis of PUMA expression. The representative image is shown. (**D**) IPZ (10 μM) was added to the culture medium 1 h before X-ray irradiation. After 20 h-incubation, the cell-culture-conditioned medium was replaced with a fresh medium, and the cells were further cultured for 3 days. After culturing, the A549 cells were harvested for apoptosis assay using annexin V/PI staining. The representative cytograms of annexin V/PI are shown. Inset numbers indicate the proportion of annexin V^+^/PI^−^ cells or annexin V^+^/PI^+^ cells. (**E**) IPZ (10 μM) was added to the culture medium 1 h before X-ray irradiation. After 20 h-incubation, the cell-culture-conditioned medium was replaced with fresh media and the cells were further cultured until the colony was observed. The surviving fraction of A549 cells is shown. Data are presented as the mean ± SD of three independent experiments performed in triplicate. (**F**) Correlation analysis was performed based on Spearman r and p values between TP63 and PUMA from the TCGA datasets.

**Figure 5 cancers-12-00908-f005:**
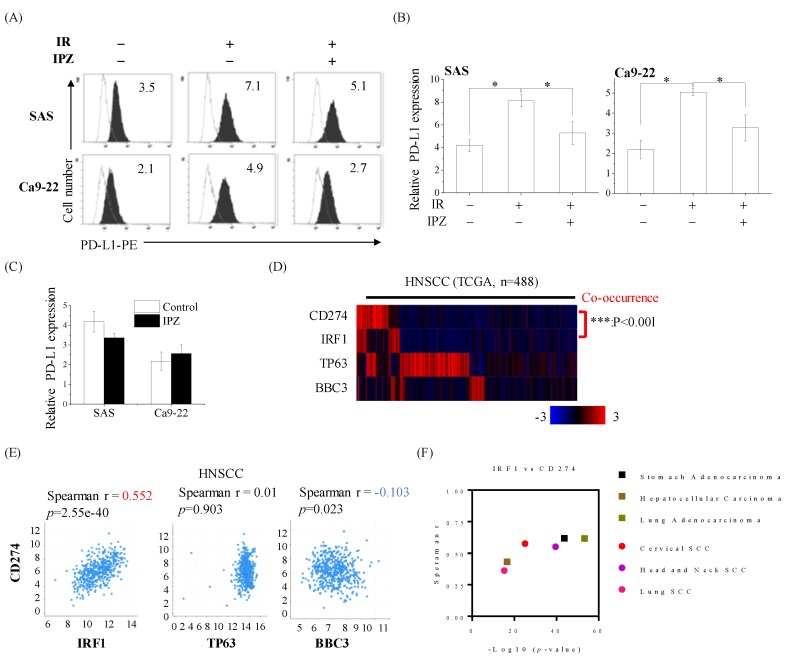
The effects of KPNB1 inhibition on radiation-increased cell surface PD-L1 expression in HNSCC cells. (**A**,**B**) IPZ (10 μM) was added to the culture medium 1 h before X-ray irradiation. The SAS cells were cultured for 4 days in the presence of IPZ after irradiation. In terms of Ca9-22 cells, the cell culture-conditioned medium was replaced with fresh media at 20 h after irradiation and the cells were further cultured for 3 days. (**A**) The representative histograms of PD-L1 expression on SAS and Ca9-22 cells are shown. The dotted line and filled black histograms indicate the isotype control and the PD-L1 expression, respectively. Inset numbers indicate the relative values of the mean fluorescence intensity of PD-L1 compared with isotype control. (**B**) Data are presented as the mean ± SD of three independent experiments. * indicates *p* < 0.01. (**C**) The quantification of PD-L1 expression levels in HNSCC cells upon IPZ treatment is shown. Data are presented as the mean ± SD of three independent experiments. (**D**) The heat map showing co-occurrence between PD-L1 (CD274) expression and IRF1. The samples were divided according to mRNA expression levels (mRNA expression z-Scores (RNA-Seq V2 RSEM) > mean + 1.0 SD) from the TCGA cohorts. P values are based on Fisher’s exact test. (**E**) Correlation between CD274 mRNA and each of IRF1, TP63, and BBC3 (PUMA) mRNA in HNSCC from TCGA. (**F**) Correlation analysis was performed based on Spearman r and *p* values from TCGA cohorts.

**Table 1 cancers-12-00908-t001:** Primer sequences used for qRT-PCR.

Sequence (5′→3′)	
PUMA F	GACGACCTCAACGCACAGTA
PUMA R	CACCTAATTGGGCTCCATCT
GAPDH F	GTCAGTGGTGGACCTGACCT
GAPDH R	AGGGGTCTACATGGCAACTG

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
