# Peer review of "Karyopherin-β1 Regulates Radioresistance and Radiation-Increased Programmed Death-Ligand 1 Expression in Human Head and Neck Squamous Cell Carcinoma Cell Lines"

_cancers, 2020, doi:10.3390/cancers12040908_

Round 1

Reviewer 1 Report

The author concluded that combination therapy of KPNB1 inhibition and radiation is promising strategy for patients with HNSCC. Study subject is important and timely. Manuscript is well readable. However, following points should be addressed to be acceptable for this Journal.

Major concerns
1) Figure 5: How contribute deltaNp63 and PUMA on the expression of PD-L1 expression. This Figure only showed parallel data.
2) Figure 3 (Line 250-262): TP63 contains two different promoters to drive two distinct isoform such as TP63 and deltaNp63. Here, it is unclear what relevance with PUMA between TP63 and deltaNp63. Is there no involvement of p53-PUMA axis?

Other concerns:
1) Basic information about PUMA and deltaNp63 should be described concisely in Introduction. Part of Discussion may be moved to Introduction.
2) Please show IC50 and treated dose of IPZ for A549 cells (Line 271, Figure 4).
3) Materials and Methods: Please indicate source of RERF-LC-MS and H1299 cell lines and their culture conditions.

Author Response

We are very glad that the reviewer feels that the study is important. We appreciate the reviewer’s comments and have tried our best to address the reviewer’s queries. All the changes have been highlighted in the revised manuscript. 

Reviewer 2 Report

This is an interesting manuscript describing the effects of KPNB1 inhibitor, IPZ, or KPNB1 knockdown on radiation response in HNSCC cell lines, SAS and Ca9-22. The study showed that IPZ enhanced radiation induced apoptosis and radiosensitivity in HNSCC cell lines but not HUVEC cells or A549 cells.  In addition, IPZ increased PUMA expression and IPZ or KPNB1 knockdown suppressed radiation induced upregulation of PD-L1 expression.

In general, the study seems to have been well planned and executed. However, a number of improvements / clarifications are required.

The authors show no significant effects in A549 cells, is this because these are derived from adenocarcinoma, why not try a lung SCC derived cell line before concluding this is unique to HNSCC?

Did the authors carry out a clonogenic assay on cells after KPNB1 knockdown? If not, please include an explanation of why this was not included.

More information on radiation dosimetry should be included.

In Figure 1D and F, a plot of surviving fraction should also be included

Figure 1 legend needs to be clearer. Each panel should be described in turn, individually.

It should be clear in the title that this is an in vitro study, ie. not in HNSCC patients but in cell lines

The standard of English needs to be improved throughout the manuscript.

Author Response

We are glad that the reviewer expressed that our study is well-planned and -executed. We appreciate the reviewer’s suggestions and have tried our best to address the reviewer’s comments. All the revisions have been highlighted in the revised manuscript.

Round 2

Reviewer 1 Report

The author has addressed all of my concerns.

Reviewer 2 Report

Thank you to the authors for addressing my comments.